# HcaNet: Haze-concentration-aware Network for Real-scene Dehazing with Codebook Priors

### Yi Liu
Wuhan University of Technology
Wuhan, China
Engineering Research Center of
Intelligent Service Technology for
Digital Publishing, Ministry of
Education
Wuhan, China
lyear@whut.edu.cn

### Jiachen Li
Wuhan University of Technology
Wuhan, China
Engineering Research Center of
Intelligent Service Technology for
Digital Publishing, Ministry of
Education
Wuhan, China
lijiachen@whut.edu.cn

### Yanchun Ma*
Wuhan Vocational College of
Software and Engineering
Wuhan, China
mayanchun@whvcse.edu.cn

### Qing Xie
Wuhan University of Technology
Wuhan, China
Engineering Research Center of
Intelligent Service Technology for
Digital Publishing, Ministry of
Education
Wuhan, China
felixxq@whut.edu.cn

### Yongjian Liu
Wuhan University of Technology
Wuhan, China
Engineering Research Center of
Intelligent Service Technology for
Digital Publishing, Ministry of
Education
Wuhan, China
liuyj@whut.edu.cn

## Abstract

In the task of image dehazing, it has been proven that high-quality codebook priors can be used to compensate for the distribution differences between real-world hazy images and synthetic hazy images, thereby helping the model improve its performance. However, because the concentration and distribution of haze in the image are irregular, the manners those simply replacing or blending the prior information in the codebook with the original image features are inconsistent with this irregularity, which leads to a non-ideal dehazing performance. To this end, we propose a haze concentration aware network (HcaNet), its haze-concentration-aware module (HcaM) can reduce the information loss in the vector quantization stage and achieve an adaptive domain transfer for regions with different degrees of degradation. To further capture the detailed texture information, we develop a frequency selective fusion module (FSFM) to facilitate the transmission of shallow information retained in haze areas to deeper layers, thereby enhancing the fusion with high-quality feature priors. Extensive evaluations demonstrate that the proposed model can be merely trained on synthetic hazy-clean pairs and effectively generalize to real-world data. Several experimental results confirm that the proposed dehazing model

outperforms state-of-the-art methods significantly on real-world images.

## CCS Concepts

• **Computing methodologies** → **Artificial intelligence**; **Computer vision problems**; **Reconstruction**.

## Keywords

Image Dehazing, Real-scene Image Dehazing, Vector Quantization

**ACM Reference Format:**
Yi Liu, Jiachen Li, Yanchun Ma, Qing Xie, and Yongjian Liu. 2024. HcaNet: Haze-concentration-aware Network for Real-scene Dehazing with Codebook Priors. In *Proceedings of the 32nd ACM International Conference on Multimedia (MM '24), October 28-November 1, 2024, Melbourne, VIC, Australia.* ACM, New York, NY, USA, 9 pages. https://doi.org/10.1145/3664647.3681314

## 1 Introduction

Single image dehazing aims to enhance visual quality by recovering a clear image from a hazy input. The task of image dehazing has diverse practical applications, particularly in areas such as traffic monitoring, autonomous driving, and terrain surveying. Dehazed images can serve as high-quality inputs for downstream tasks such as object detection and tracking in haze-affected environments.

In recent years, deep learning-based dehazing methods have demonstrated exceptional performance and emerged as the predominant approaches. Early DNN-based approaches [5, 23, 30] primarily employ deep networks for estimating physical parameters and then use scattering model [6] to derive haze-free images. Subsequently, researchers are increasingly inclined to employ DNNs to directly restore the haze-free image in order to mitigate cumulative errors arising from parameter estimation [10, 29, 33]. While these

---

*Corresponding author

methods demonstrate excellent performance on synthetic datasets, there is a serious performance degradation when these methods are transferred directly to real-world datasets. Other methods, such as PSD [9] and RIDCP [38], are specifically designed for real scenes. For instance, PSD [9] incorporates physical priors manually and fine-tunes the model by utilizing the real-world data in an unsupervised manner, RIDCP [38] further utilizes high-quality codebook priors to address unique ill-posed problems those are unique to real-world datasets.

According to the aforementioned works, the codebook is capable of serving as supplementary high-quality prior information, thus helping exhibit excellent performance in restoring degraded images in real-world scenarios. However, simply replacing or blending the prior information in the codebook with the original image features may lead to distortion and loss of details in the dehazed images. This is because the concentration and distribution of haze in the image are irregular, while the above simple manner of prior information introduction is inconsistent with this irregularity.

To address the aforementioned issues, we propose a dehazing network that is aware of haze-concentration, referred as HcaNet (Haze-concentration-aware Network for Real-scene Dehazing with Codebook priors). The goal of this network is to effectively integrate high-quality feature priors, while simultaneously fusing the retained degradation feature in the corresponding areas according to their haze concentration, to handle diverse real-world haze scenarios. Specifically, we construct a reconstruction-oriented dictionary called high-quality codebook by training an image restoration network on numerous high-quality images. This high-quality codebook serves as a valuable resource containing abundant high-quality feature priors that facilitate the transformation of features from hazy regions into clean domains. Moreover, unlike previous approaches that arbitrarily employ vector quantization (VQ) [16, 37, 38, 42] for feature replacement or blending, we specifically design a haze-concentration-aware module (HcaM) for fusing features derived from both high-quality priors and hazy areas. This module not only minimizes information loss but also enables adaptive domain transfer for regions with varying degrees of degradation.

For the recovery of fine-grained details, we further propose a frequency selective fusion module (FSFM) to refine the texture of the reconstruction stage features, which can facilitate the transmission of shallow information retained in haze areas to deeper layers, thereby enhancing the utilization of low-quality features similar to the widely employed U-Net [31] for low-level visual tasks [29, 33, 34]. However, in contrast to the direct jump link employed by U-Net, our approach selectively preserves solely high-frequency information of the shallow features, which is in consideration of the fact that the interference of haze as a kind of low-frequency information will directly affect the recovery of texture information. Subsequently, we fuse the high-frequency information with high-quality feature priors to enhance the output image quality.

In summary, our main contributions include:

- To cope with the inconsistency caused by the haze concentration and effectively utilize the high-quality prior information, we propose a novel haze-concentration-aware module capable of effectively integrating low-quality features with high-quality codebook features for regions exhibiting varying degrees of degradation.

- To enhance the utilization of information on degraded areas, we propose a frequency selective fusion module that facilitates the transmission of shallow information to deeper layers. Subsequently, this information is selectively integrated with high-quality feature priors to augment the model's ability in recovering fine textures.
- We expand the dehazing task to real-world scenarios, specifically blind dehazing, by training on synthetic data and directly transferring to real haze situations. Our model exhibits exceptional generalization performance when applied to real data.

This paper is organized as follows: the related works are reviewed in the next section, and Section 3 explains the model details of the HcaNet. The experimental setup, experimental results, and ablation experiments are introduced in Section 4. In the last section, we give the conclusion of this work.

## 2 Related Work

***Image Dehazing.*** Early dehazing methods depend on the physical scattering model [6] and usually remove the haze using handcraft priors from empirical observation. He et al. [18] use the dark channel prior (DCP) to estimate the transmission map, assuming that the lowest pixel value should be close to zero in at least one channel except for the sky region. Zhu et al. [43] propose a simple but powerful color attenuation prior (CAP) that creates a linear model for modeling the depth of field of hazy images, thus effectively dehazing from a single image. However, these hand-crafted priors are not always reliably estimated in real scenarios, and therefore cause unsatisfactory results.

With the general success of deep learning in image processing tasks, data-driven dehazing methods have made great progress. Some early approaches use convolutional neural networks (CNNs) to estimate the transmission map and atmospheric light in the physical scattering model [5, 30, 41]. To avoid cumulative errors in parameter estimation, many works use end-to-end approach to directly estimate haze-free images [8, 12, 17]. These learning-based methods have achieved significant performance on synthetic datasets, nevertheless, their performance degrades significantly once they are transferred directly to real-world datasets.

***Real Scene Image Dehazing.*** Some recent works also focus on real scenes of dehazing. Li et al. [25] propose a semi-supervised approach to training neural networks on real datasets, utilizing a loss function informed by prior knowledge. PSD [9] uses a dehazing model pre-trained on synthetic paired data, and then fine-tunes the model in an unsupervised manner on real-world data via the proposed prior loss committee. However, the direct use of these hand-crafted priors for transferring synthetic to real domains does not address the shortcomings of these priors themselves. There are also approaches that utilize generative models like GANs [14] to enhance the generalization ability of models in real-world scenarios. These works use GANs to generate hazy images that are closer to the real domain to achieve domain translation. Yang et al. [39] propose a self-augmented image dehazing framework, termed D4. It is capable of re-rendering hazy images with different thicknesses which further benefits the training of the dehazing network. Nevertheless, GANs are easy to generate artifacts within their outputs,

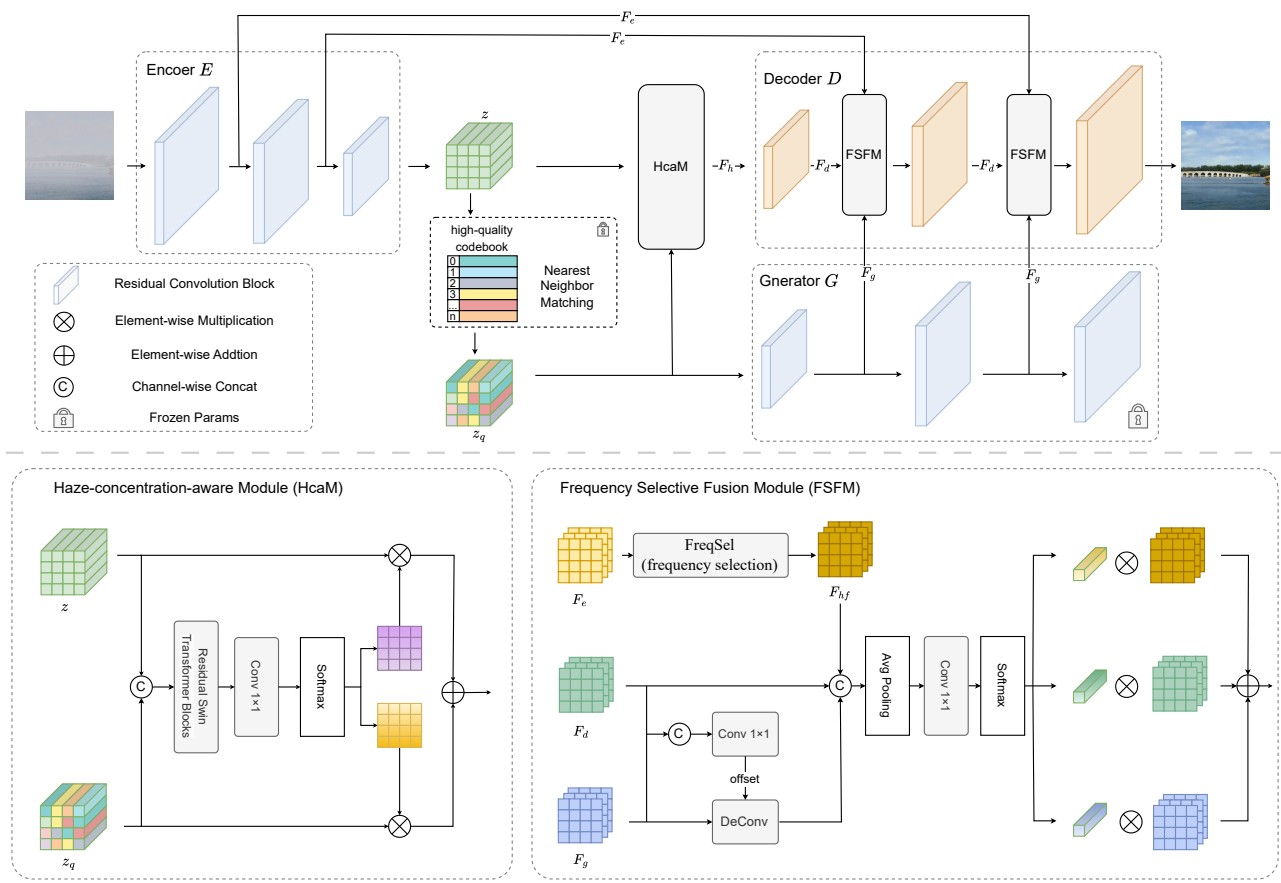

**Figure 1: Overview of HcaNet. The encoder $E$ maps the degraded image to the latent space and then uses nearest neighbour lookup in codebook to get the high-quality priors. HcaM fuse the degraded features with high-quality priors adaptively. The pre-trained generator $G$ is responsible for improving the high-quality codebook priors, while the decoder $D$ recovers a clean image by flexibly utilising the priors and shallow information through the FSFM.**

which can detrimentally affect the training of models. RIDCP [38] presents a paradigm for real image dehazing from the perspectives of synthesizing more realistic hazy data and introducing more robust codebook priors into the network. However, codebook priors are themselves a form of noise, and their direct introduction can also lead to some loss of structural and detailed information.

***Codebook Learning.*** VQVAE [36] first introduces a generative autoencoder model that learns discrete latent representations, also known as "codebook". The following VQGAN [13] employs perceptual and adversarial loss to train the visual codebook, resulting in better image generation quality with a relatively small codebook size. The representation dictionary-based generative models inspire various impressive image generation works [7, 15, 40]. In recent studies, codebook-based methods explore the use of learned high-quality dictionary or codebook that contain more generic and detailed information for image restoration. CodeFormer [42] employs a transformer to establish the appropriate mapping between

low-quality features and code indices. Subsequently, it uses the code index to retrieve the corresponding feature in the codebook for image restoration. RestoreFormer [37] and VQFR [16] attempt to directly incorporate low-quality information with the codebook information based on the codebook priors. However, these methods may encounter severe degradation limitations, as the low-quality information can negatively impact the high-quality information derived from the codebook.

## 3 Methodology

The primary objective of real-scene image dehazing is to alleviate the domain gap and mitigate information loss that arises during the restoration process from hazy images to clean images. The challenge lies in achieving varying levels of restoration for areas with different degrees of degradation, which significantly impacts the accuracy and effectiveness of the restoration process.

To cope with the above uncertainty of real scene dehazing, we propose leveraging codebook priors and designing the HcaM for the

different distribution of haze in different regions to achieve adaptive domain transformation. Additionally, in order to further exploit the feature information of the degraded region, we propose the FSFM to facilitate the transmission of shallow high-frequency information to deeper layers, thereby enhancing the decoder's ability to generate higher-quality images.

The whole pipeline is shown in Figure 1. An encoder $E$ is first deployed to extract representation $z$ of the degraded low-quality image and its nearest high-quality priors $z_q$ are fetched from the high-quality codebook. Then, the proposed HcaM performs adaptive fusion of $z$ and $z_q$ for dynamic domain transfer (Sec 3.2). Finally a decoder $D$ is applied on the fused representation $F_h$ to restore a clean image. Furthermore, in order to utilize the information retained in haze areas and further recover more detailed information, at each stage of the reconstruction we complementarily employ the FSFM to fuse the high-frequency information from the encoder as well as the high-quality priors from the pre-trained generator $G$ (Sec 3.3).

## 3.1 High-quality Codebook Priors Construction

In order to handle complex real-world scenarios, we propose the deployment of codebook priors, which encompass a diverse range of high-quality features that are instrumental in facilitating the transfer of input images from the hazy domain to the clear domain. In this subsection, we initially present an introduction to the generation process of the high-quality codebook. Given a clean image patch $x \in \mathbb{R}^{H \times W \times 3}$ is first passed through the encoder $E$ to produce its output feature $z = E(x) \in \mathbb{R}^{h \times w \times n_z}$ , where $n_z$ is the dimension of latent vectors. Then the vector quantized representation of $z_q$ is calculated by finding the nearest neighbours of each element $z_i \in \mathbb{R}^{n_z}$, in the high-quality codebook $\mathbb{C} = \{z_i\}_{i=0}^{K}$ with $K$ discrete codes as follows:

$$z_q = \mathbf{q}(z) := \left( \arg \min_{z_i \in \mathbb{C}} \|z - z_i\| \right) \in \mathbb{R}^{h \times w \times n_z}. \tag{1}$$

where $\mathbf{q}(\cdot)$ denotes the element-wise quantization. The Generator $G$ maps the quantized representation $z_q$ back to the RGB space. The overall reconstruction mechanism can be formulated as:

$$\hat{x} = G(z_{\mathbf{q}}) = G(\mathbf{q}(E(x)) \approx x \tag{2}$$

$\hat{x}$ is the reconstructed result, which should be as close as possible to the input clean image $x$.

The elements $z_i$ in $\mathbb{C}$ are randomly initialized by a uniform distribution. For updating them to capture high-quality information, we follow the previous works on vector quantization [13, 36, 42] by simply copying the gradients from $G$ to $E$ for backpropagation. The code-level loss is formulated as:

$$\mathcal{L}_{code} = \|\text{sg}(z) - z_q\|_2^2 + \beta \|z - \text{sg}(z_q)\|_2^2, \tag{3}$$

where sg[·] denotes the stop-gradient operation and and $\beta = 0.25$ is a hyper-parameter to control the update frequency of the codebook. The loss can make $z$ close to $z_q$ which is extracted from high-quality undegraded images, $z_q$ contains high-quality information which can benefit image restoration. Besides the code-level loss, we also adopt four image-level reconstruction losses to improve the quality of the final reconstructed result: L1 loss $\mathcal{L}_{l1}$ for basic pixel reconstruction, perceptual loss $\mathcal{L}_{per}$ [20] for perceptual quality,

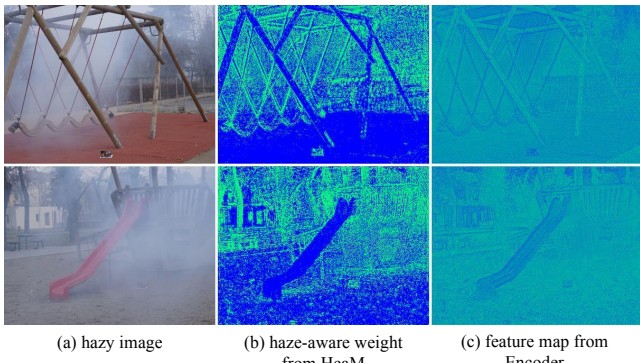

(a) hazy image     (b) haze-aware weight     (c) feature map from
               from HcaM             Encoder

**Figure 2: visualization of weight map from the HcaM.**

adversarial loss $\mathcal{L}_{adv}$ for texture generation, and semantic loss $\mathcal{L}_{sem}$ to encourage the texture to be conditioned on semantics. These losses are denoted as:

$$\mathcal{L}_{l1} = \|x - \hat{x}\|_1; \mathcal{L}_{sem} = \|Conv(z_q) - \phi(x)\|_2^2; \tag{4}$$
$$\mathcal{L}_{adv} = [\log D(x) + \log(1 - D(\hat{x}))].$$

where $\phi$ denotes the commonly used feature extractor, such as the VGG19 [32] and ResNet [19], $Conv(\cdot)$ represents the convolutional layer. Finally, The complete objective of high-quality codebook learning $\mathcal{L}_{Dict}$ is:

$$\mathcal{L}_{Dict} = \mathcal{L}_{code} + \mathcal{L}_{l1} + \mathcal{L}_{per} + \lambda \mathcal{L}_{sem} + \lambda \mathcal{L}_{adv}, \tag{5}$$

where $\lambda$ is trade-off weight that is set to 0.1.

## 3.2 Haze-concentration-aware Module (HcaM)

Previously, we introduced the generation of codebook priors, the encoder $E$ is able to map the input image to latent space and the generator $G$ is capable of mapping the priors from the codebook to RGB space. In the dehazing phase, we continue to use the encoder $E$ trained in the previous phase and use a parallel decoder architecture, where the generator $G$ is responsible for generating high-quality prior features and the decoder $D$ in the main branch is used for decoding the features after adaptive fusion and reconstructing clear images. This is designed to allow the model to extract cleaner features and make better use of the codebook priors.

***HcaM Construction.*** Similar to other methods using vector quantization, for the features extracted by the encoder $E$, we use nearest neighbor matching directly in the high-quality codebook to obtain the high-quality feature representation $z_q$. Such direct substitution, although it can help degraded feature $z$ to jump to the clear domain, also suffers from information loss. For a hazy image, not every region in the image is hazy. From the code point of view, not every code of feature $z$ needs to jump to the clear domain. To solve this problem and combine it with practical scenarios, we propose the Haze-concentration-aware Module to reduce information loss, achieving adaptive domain transfer for areas with varying levels of degradation. Specifically, we first connect $z$ and $z_q$ in the channel dimension and then feed them into a series of Residual Swin Transformer Blocks (RSTBs), which are better able to capture long-range dependencies. We then apply a $1 \times 1$ convolutional layer

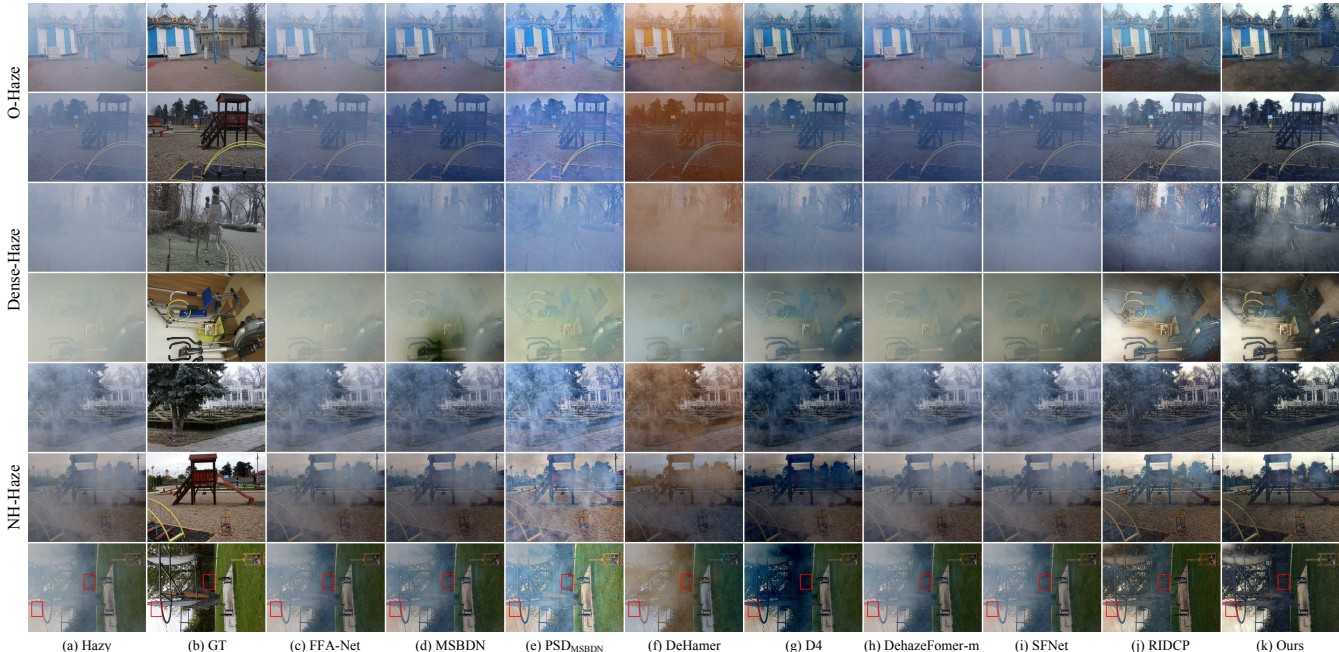

**Figure 3: Qualitative comparison of dehazed images with different methods. Visualizations are displayed sequentially with the O-Haze [4] dataset in the 1st and 2nd rows, Dense-Haze [2] dataset in the 3rd and 4th rows, and NH-Haze [3] dataset in the 5th, 6th and 7th rows. In the 7th row, we box out some areas and show them enlarged in Figure 4.**

followed by a Softmax activation function to get our haze-aware weights $w_h \in \mathbb{R}^{h \times w \times 2}$. Finally, the feature $F_h$ after adaptive fusion is summed by multiplying $z$ and $z_q$ with their corresponding haze-aware weights. The whole process can be described as

$$w_{h1}, w_{h2} = Softmax(Conv_{1\times1}(RSTBs(Concat(z, z_q))))$$
$$F_h = z \times w_{h1} + z_q \times w_{h2} \tag{6}$$

where the $Concat(\cdot)$ refers to the concatenation operation, the $Conv_{1\times1}(\cdot)$ represents the convolutional layer and the $Softmax(\cdot)$ means the Softmax activation layer.

To verify the sensitivity of HcaM to haze information, we visualised the haze-aware weight $w_{h2}$ and degradation feature $z$ as depicted in Figure 2. It is evident that despite the uneven distribution of haze in the input image, the features extracted by the encoder are still unable to effectively distinguish these regions. According to the visualization of haze-aware weights, the brighter colors in the figure are highly consistent with the haze regions. This indicates that our HcaM is capable of properly identifying the haze regions and providing an effective guide for introducing high-quality features.

### 3.3 Frequency Selective Fusion Module (FSFM)

To further mitigate texture defects in the results, we further propose a frequency selective fusion module (FSFM) to refine the texture features in the reconstruction phase through shallow high-frequency features and high-quality priors simultaneous guidance. As shown in Figure 1, we use a parallel decoder architecture, where the pre-trained generator $G$ is responsible for generating high-quality prior features and the decoder $D$ in the main branch is responsible for

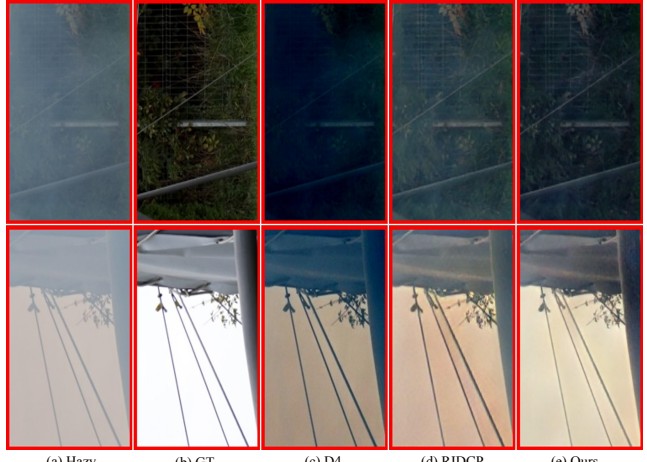

**Figure 4: Detailed visualization comparison of some methods on the NH-Haze [3] dataset. We enlarge the boxed out area in the 7th row of Figure 3 to get a better display of the details.**

decoding the features after adaptive fusion and reconstructing clear images. The interaction of features is realized through FSFM at each layer of decoding. Specifically, at layer $i$, we use deformable convolution [11] to align feature $F_g^i$ in $G$ with feature $F_d^i$ in $D$, which can be formulated as:

$$O^i = Conv_{1\times1}(Concat(F_d^i, F_g^i))$$
$$F_w^i = DeConv(F_g^i, O^i) \tag{7}$$

**Table 1: Quantitative comparison on three paired real-world datasets, red and blue indicate the best and the second-best, respectively. The last row is a comparison of our results with the second best.**

| method | venue | O-Haze | | Dense-Haze | | NH-Haze | |
|---|---|---|---|---|---|---|---|
| | | SSIM↑ | PSNR↑ | SSIM↑ | PSNR↑ | SSIM↑ | PSNR↑ |
| FFA-Net [29] | AAAI2020 | 0.651 | 14.737 | 0.450 | 10.360 | 0.481 | 11.938 |
| MSBDN [12] | CVPR2020 | 0.696 | 16.626 | 0.464 | 11.025 | 0.502 | 12.408 |
| $PSD_{MSBDN}$ [9] | CVPR2021 | 0.653 | 11.675 | 0.447 | 9.736 | 0.543 | 10.601 |
| DeHamer [17] | CVPR2022 | 0.692 | 15.980 | 0.459 | 11.088 | 0.493 | 12.221 |
| D4 [39] | CVPR2022 | 0.701 | 16.670 | 0.440 | 11.328 | 0.508 | 12.783 |
| DehazeFomer-m [33] | TIP2023 | 0.684 | 16.044 | 0.464 | 10.945 | 0.486 | 12.001 |
| SFNet [10] | ICLR2023 | 0.660 | 15.130 | 0.463 | 11.010 | 0.490 | 12.140 |
| RIDCP [38] | CVPR2023 | 0.696 | 16.005 | 0.462 | 10.621 | 0.527 | 12.866 |
| HcaNet (Ours) | - | 0.733 | 16.778 | 0.497 | 11.410 | 0.564 | 13.156 |

where $O^i$ denotes the offsets for deformable convolution and $DeConv$ is the deformable convolutional layer. $F_w^i$ is the features after warping. For the shallow features of jump connections, we apply frequency selection to fully extract the useful frequency information in them. More specifically, inspired by previous work related to the use of frequency information [10], we use a set of learned high-pass filters to generate high-frequency subbands. The learned filters are shared across the group dimension to strike a balance between complexity and feature diversity. The process can be formulated as:

$$F_{hf}^i = Conv_{1\times1}(FreqSel(F_e^i)) \qquad (8)$$

where $F_{hf}^i$ is the high-frequency information from the input features and $FreqSel$ refers the frequency selection module. We then connect the three features in the channel dimension and send them to a pooling layer and a $1 \times 1$ convolutional layer, and finally generate their respective weights through the Softmax activation layer in order to thoroughly fuse the information. The procedure can be expressed as:

$$w_d^i, w_g^i, w_{hf}^i = Softmax(Conv_{(1\times1)}(Concat(F_d^i, F_w^i, F_{hf}^i)))$$
$$F_d^{i+1} = F_d^i \times w_d^i + F_w^i \times w_g^i + F_{hf}^i \times w_{hf}^i \qquad (9)$$

## 4 Experiment

In this section, we conduct a series of experiments to evaluate the performance of our model in real-world dehazing scenarios. Firstly, we present a comprehensive experimental setup and subsequently compare our model with several state-of-the-art dehazing methods across multiple datasets. Furthermore, an ablation study is conducted to investigate the impact of the designed components on the behavior of our dehazing model.

### 4.1 Experimental Settings

*Pre-training.* In order to obtain high-quality priors, we first train the codebook on DIV2K [1] and Flickr2K [27] datasets, those contain high-resolution and texture-sharp images and widely used for high-quality reconstruction tasks [7, 26, 28].

*Training.* Following convention, we train our model on Outdoor Training Set (OTS) of REalistic Single Image DEhazing (RESIDE) [24]. Specifically, we employ data augmentation techniques

**Table 2: Quantitative comparison on RTTS [24] datasets, red and blue indicate the best and the second-best, respectively.**

| Method | BRISQUE↓ | NIMA↑ | MUSIQ↑ |
|---|---|---|---|
| Hazy image | 37.011 | 4.325 | 53.766 |
| DCP [18] | 32.448 | 4.523 | 52.164 |
| CAP [43] | 35.072 | 4.502 | 50.179 |
| MSBDN [12] | 28.743 | 4.140 | 53.727 |
| DeHamer [17] | 33.866 | 3.866 | 52.334 |
| D4 [39] | 33.206 | 3.723 | 53.039 |
| DehazeFomer-m [33] | 34.039 | 4.582 | 54.013 |
| SFNet [10] | 36.032 | 4.583 | 54.146 |
| RIDCP [38] | 18.782 | 4.427 | 55.627 |
| HcaNet (Ours) | 17.276 | 4.888 | 58.263 |

during the training phase to incorporate additional factors such as low light and blurring into the degraded images, with the aim of enhancing the model's capacity to adapt more effectively to real-world environments.

*Evaluating.* 4 real-world datasets O-Haze [4], Dense-Haze [2], NH-Haze [3] and RTTS [24] are used to evaluate the models. The first 3 datasets have hazy-clean pairs while RTTS only has hazy images. In order to evaluate the generalization capacity of our dehazing model to realistic scenarios with more challenges, we train it merely on synthetic data, and directly evaluate it on the aforementioned 4 test sets without any fine-tuning or re-training.

*Evaluation Metrics.* We use both reference and non-reference metrics to comprehensively evaluate the performance of the dehazing model. For reference metrics, PSNR and SSIM are used to measure the distortion between the dehazed image and the real image in terms of pixels and structures, respectively. For non-reference metrics, NIMA [35] and MUSIQ [21] are used to measure how much the dehazed images are similar to natural images in terms of statistical regularities and granularities.

### 4.2 Implementation Details

During training, we randomly resize and crop the input to the size $256 \times 256$ and flip it with half probability. For all stages of training,

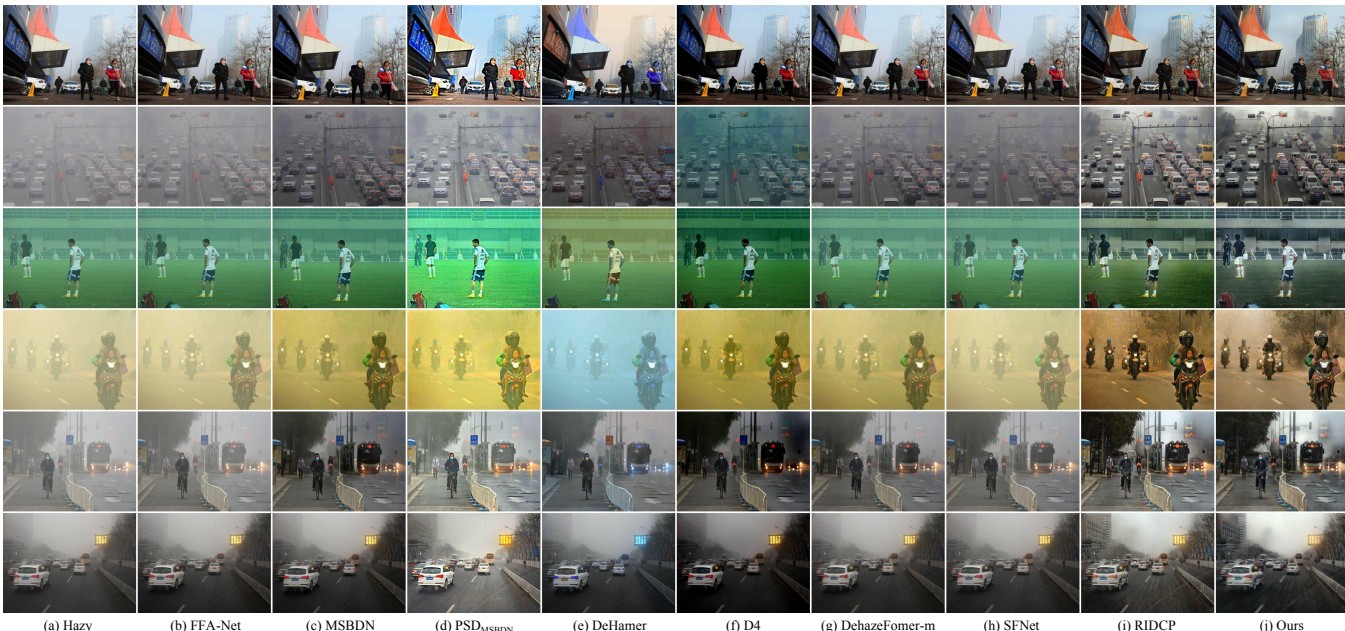

| (a) Hazy | (b) FFA-Net | (c) MSBDN | (d) PSD$_{MSBDN}$ | (e) DeHamer | (f) D4 | (g) DehazeFomer-m | (h) SFNet | (i) RIDCP | (j) Ours |

**Figure 5: Visual comparison on RTTS [24] dataset.**

**Table 3: Ablation studies of HcaM and FSFM, red and blue indicate the best and the second-best, respectively.**

| setting | Model | | O-Haze | | Dense-Haze | | NH-Haze | |
|---------|-------|------|--------|------|------------|------|---------|------|
|         | HcaM | FSFM | SSIM↑ | PSNR↑ | SSIM↑ | PSNR↑ | SSIM↑ | PSNR↑ |
| i   |     |     | 0.696 | 16.005 | 0.462 | 10.621 | 0.527 | 12.866 |
| ii  | ✓   |     | 0.703 | 15.611 | 0.491 | 10.958 | 0.532 | 12.471 |
| iii | ✓   | ✓   | 0.733 | 16.778 | 0.497 | 11.410 | 0.564 | 13.156 |

we use Adam [22] optimizer with batch size as 2, the learning rate is fixed to 0.0001. In pre-training stage, the encoder $E$ and generator $G$ are trained for 500k iterations on DIV2K [1] and Flickr2K [27]. In the training stage, the model is trained on synthetic data for 50k iterations. Our model is implemented with the PyTorch framework and trained by one NVIDIA RTX 3090 GPU.

### 4.3 Performance Comparisons

***Quantitative Comparison.*** We conducted a comprehensive performance evaluation of our proposed method against several state-of-the-art dehazing approaches trained based on synthetic data, 1) including FFA-Net [29], MSBDN [12], SFNet [10], and Transformers such as DehazeFormer-m [33]. 2) Additionally, we compare our method to several leading real scene dehazing methods such as PSD [9], D4 [39], and RIDCP [38]. 3) We also compare our model with some traditional methods based on priors, such as DCP [18] and CAP [43].

As presented in Table 1, our method significantly outperforms other baselines across all the paired real-world datasets. Some methods that have been proved to perform well on synthetic datasets, such as SFNet [10] and DehazeFormer-m [33], have significantly degraded performance when transferred to real-world datasets,

mainly because of the large domain gap between synthetic and real data. Our method performs better than some real-scene dehzing methods in terms of detail and structure mainly due to the two modules we propose. The HcaM can achieve domain transformation while reducing information loss, while FSFM introduces more high-frequency information. As a result, our method achieves better results in terms of reference metrics.

Table 2 shows the evaluation results on real-world hazy images in RTTS dataset which lacks of corresponding clean images. Non-reference metrics are used to evaluate the various methods. Results demonstrate that our method is capable of outputting higher perceptual quality dehazing results, which proves the effectiveness of our method in real-world scene dehazing. It is worth noting that while traditional methods based on priors outperform some deep learning based methods on a few metrics, their performance remains relatively average and there is still a notable gap compared to our approach.

***Qualitative Comparison.*** We perform the qualitative comparison on various datasets. As depicted in Figure 3, it is evident that the D4 [39] and Dehamer [17] are significantly impacted by haze and the background color of the images, resulting in a relatively large color deviation. Methods such as SFNet [10], Dehazeformer-m

**Table 4: Ablation studies of FSFM design, red and blue indicate the best and the second-best, respectively.**

| setting | Model | O-Haze | | Dense-Haze | | NH-Haze | |
|---------|-------|--------|--------|------------|--------|---------|--------|
| | | SSIM↑ | PSNR↑ | SSIM↑ | PSNR↑ | SSIM↑ | PSNR↑ |
| i | w/o skip | 0.695 | 13.368 | 0.454 | 9.432 | 0.569 | 11.536 |
| ii | w/o *FreqSel* | 0.676 | 15.251 | 0.474 | 10.960 | 0.501 | 12.399 |
| iii | MHCA | 0.682 | 15.826 | 0.464 | 11.037 | 0.520 | 12.703 |
| iv | Ours | 0.733 | 16.778 | 0.497 | 11.410 | 0.564 | 13.156 |

[33], and FFA-Net [29] are not effective in removing dense haze scenarios and often leave behind a significant amount of haze residue, resulting in blurred scenes. In comparison to the above methods, the RIDCP [38] method significantly enhances overall dehazing effects and color restoration after dehazing. However, in some local areas with high haze concentration, there still remains an issue of imperfect detail restoration. Compared to these methods, our method exhibits minimal haze residue in challenging dense haze scenes. Although our model is not yet capable of completely eliminating the haze, it still allows for a relatively clear reconstruction of the entire scene.

In terms of detailed comparison, we chose to compare our method with the two best-performing methods D4 [39] and RIDCP [38]. The local zoom parts are taken from the last row in Figure 3 and we have highlighted them with 2 red rectangular boxes in each of the hazy images. As illustrated in Figure 4, while D4 [39] and RIDCP [38] are capable of removing most of the haze from the overall view, their performance in preserving local details is still unsatisfactory. The results show more serious degradation in reconstruction, with the lines under the cover of haze appearing to have a blurred structure. The 2nd row shows that there is even a slight color deviation in RIDCP. Instead, our method reconstructs better in heavily degraded areas because it takes into account the high-frequency information in the degraded areas.

Comparisons on RTTS [24] dataset are shown in Figure 5, our model removes the overall haze, while being closer to reality in color reproduction. On the whole, our approach produced the best perceptual results in terms of brightness, colorfulness, and haze residue compared to others.

## 4.4 Ablation Study

***Effectiveness of HcaM and FSFM..*** Ablation experiment results of the proposed HcaM and FSFM are reported in Table 3. Specifically, we remove HcaM and FSFM separately to assess their contributions. Results show that both HcaM and FSFM can bring some performance improvement, which further validates the effectiveness of our design.

The item-(ii) verifies that high-quality feature priors can help the model to achieve domain transformation, and our proposed HcaM can domain translation while reducing the information loss in feature replacement. Simultaneously, it also demonstrates that the incorporation of high-quality prior information inevitably leads to information loss, primarily manifested by the decline in PSNR for some datasets.

The item-(iii), on the other hand, demonstrates that the FSFM effectively supplements the high-frequency details when utilizing

prior features, thereby achieving an optimal balance between the two metrics.

***Effectiveness of FSFM Design.*** To study the effectiveness of the proposed Frequency Selective Fusion Module, three variant settings are presented in Table 4: (i) Without skip connection (w/o skip); (ii) Direct skip connection without frequency selection (w/o *FreqSel*); (iii) Using multi-head cross-attention (MHCA) [37] to perform feature fusing. The results show that our FSFM is capable of extracting effective information from shallow features, improving the overall modeling ability of the model and the recovery of detailed texture through extensive interaction with high-quality prior features.

## 5 Conclusion

In this paper, we propose a novel dehazing network towards real-scene using codebook priors, which demonstrates superior generalization ability. We employ the codebook priors to cope with the complexity of the real scene. Moreover, to address the problem of uneven distribution of degradation in hazy images of real scenes, we designed HcaM to adaptively transfer from the hazy domain to the clear domain. Meanwhile, the proposed FSFM can further help the model to supplement more detailed information. By effectively exploiting high-quality codebook priors, our model achieves good generalization performance on several real datasets.

***Limitation and Future Work.*** Although our approach has achieved good results in real-scene dehazing, we find that there are still some difficulties that need to be solved. First of all, the problem of computational efficiency, real scene dehazing high application value, while maintaining good results to make the model more lightweight is a worthwhile research direction in the future. Furthermore, comparisons with traditional methods reveal that in some scenarios, prior-based traditional methods can even outperform learning-based approaches. Integrating more prior knowledge to adapt to more complex real-world environments is also a promising direction for future research. The use of codebook priors has already demonstrated good performance in real-world settings, and further research can focus on training richer codebooks and developing better matching rules.

## Acknowledgments

This research is partially supported by National Natural Science Foundation of China (Grant No. 62271360) and Key Research Program of Hubei (Grant No. 2023BAB085) and the open-end fund of State Key Laboratory of Advanced Design and Manufacturing for Vehicle Body (NO. 32115012).

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
