# OpenReview forum: "HcaNet: Haze-concentration-aware Network for Real-scene Dehazing with Codebook Priors"
_acmmm.org/ACMMM/2024/Conference — MM2024 Poster_

### Official Review · Reviewer_gVd3 · 2024-05-17

**Rating:** 4
**Confidence:** 4

**Summary:**

This paper proposes the following contributions:
(1) Utilize the codebook to capture the prior information of images and utilize it for real-image dehazing task. To bridge the gap caused by inconsistent haze distribution, HcaM is proposed.
(2) Proposing a frequency-selective fusion module that facilitates the transmission of shallow information to deeper layers, which boosts the reconstruction process.
(3) Experiments have demonstrated the ability of real-world dehazing.

**Strengths:**

1. The target of this work is real-world dehazing, which is a very valuable issue in the field of image processing.
2. This work has some novelty. utilizing the prior captured by the codebook and proposing a module to bridge the gap caused by inhomogeneous haze distribution is reasonable.
3. The paper is easy to follow, with great organization.
4. The paper shows promising experimental results, especially in the real-world dehazing benchmarks.

**Limitations:**

1. Some typos should be missed. For example, there should be a space between (. The authors should check the whole paper carefully.
2. The key contribution of the work is HcaM. And I have some concerns about this module :
(a) Although the author have provided some convincing visualization results in Figure 2. However, I want to see more samples (at least 5) to prove the effectiveness of HcaM. I also argue the author complement more results in Appendix.
(b) The motivation of the network design should be explained in detailed in Sec 3.2. In the output of the HcaM, it seems a 2-classification output. Is there a supervision here? What is the meaning of the wh1 and wh2?
(c) From Table 3, the effectiveness of the sole HcaM is not so significant, it should be utilized with the FSFM. The author should explain the reason.

If the concerns will be addressed, I will raise my scores. Looking forward to your reply.

**Suitability:**

3

---

### Official Review · Reviewer_Di8Z · 2024-05-21

**Rating:** 3
**Confidence:** 4

**Summary:**

This paper proposes a new dehazing method called "HcaNet" towards real-scene hazy images using codebook priors. The key ideas include: 1) Using the codebook priors to cope with the complexity of the real scene. 2) A HcaM is designed to adaptively fuse the degraded image features with the high-quality codebook priors based on the haze concentration in different regions of the image. 3) A FSFM is proposed to selectively incorporate high-frequency shallow features from the input image into deeper layers to better recover fine texture details.

**Strengths:**

1.	This paper proposed a new prior for image dehazing called high-quality codebook priors, which can facilitate the transformation of features from hazy regions into clean domains.
2.	As shown in Fig. 1, the framework of this paper is very clarity. The structures of modules are all clear and can be easily understood.
3.	The authors provide a comprehensive overview of the high-quality codebook generation process (Section 3.1), which is crucial for understanding the overall approach.
4.	The explanations of HcaM and FSFM are both adequate by using lots of formulations, such as Eqs. 6-9.
5.	The authors conducted extensive experiments to evaluate the proposed method's performance on multiple real-world dehazing datasets, including both paired (with ground truth) and unpaired scenarios.

**Limitations:**

1. Technical errors: The HcaM in Fig. 1 is wrong, in which the arrow direction on yellow block is reversed.
2. Please provide more details on the dataset used for pre-training the high-quality codebook, such as the number of images, resolution, and diversity of the images.
3. Please discuss the computational complexity and inference time of the proposed model, as well as strategies for reducing the computational burden.
4. Provide more visual examples and analyses of failure cases, highlighting scenarios where the proposed method may struggle or produce suboptimal results.
5. It’s better to discuss the limitations of the current work and outline future research directions, including potential improvements to the model architecture, training strategies, or techniques for incorporating additional prior knowledge or constraints.

**Suitability:**

2

---

### Official Review · Reviewer_a6Fu · 2024-05-22

**Rating:** 4
**Confidence:** 2

**Summary:**

A haze concentration aware network(HcaNet), its haze-concentration aware module(HcaM) can reduce the information loss in the vector quantization stage and achieve an adaptive domain transfer for regions with different degrees of degradation. The proposed algorithm's dehazing effect is significantly better than that of other algorithms.

**Strengths:**

The proposed algorithm achieves significantly better dehazing effects on real images compared to other algorithms.

**Limitations:**

1 The time complexity has not been explained, and with a training epoches of 500K, could it be a result of overfitting?

2 For a fair comparison, algorithms need to be trained on the same dataset, or the test set should significantly differ from the training sets of all algorithms.

3 We know that deep learning dehazing algorithms rely heavily on data, resulting in significant performance differences across various datasets. In contrast, traditional dehazing algorithms are more stable and less affected by data variations. The authors need to compare their method with traditional dehazing algorithms.

4 The data and distribution of the training set, test set, and validation set have not been described in detail.

**Suitability:**

2

---

### Official Review · Reviewer_5mim · 2024-05-24

**Rating:** 4
**Confidence:** 3

**Summary:**

The paper proposes a Haze-concentration-aware Network (NcaNet) for Real-scene Dehazing with Codebook Priors. NcaNet aims to cope with the inconsistency caused by the haze concentration through the use of a haze-concentration-aware module which adaptively fuses degraded feature and high-quality codes. Additionally, a frequency-selective fusion module is proposed to enhance the utilization of information on degraded areas.

**Strengths:**

1.	The main ideas of the paper are clearly presented and theoretically plausible. The Haze-concentration-aware Module fuses features derived from both high-quality priors and hazy areas to address varying degrees of degradation. The frequency-selective fusion module facilitates the transmission of shallow, high-frequency information to deeper layers.
2.	Experimental results demonstrate that the proposed NcaNet outperforms other single image dehazing methods on both synthetic dataset and real-world datasets.

**Limitations:**

1. In Figure 2, the upper part of the slide and the surrounding fence have similar haze concentrations, but their haze-aware weights differ significantly.
2. Furthermore, in Table 3, the ablation studies of HcaM reveal a significant decrease in PSNR for two datasets after applying HcaM. This raises doubts about whether the Haze-concentration-aware module is effective.

**Suitability:**

2

---

### Meta-Review · Area_Chair_xcyr · 2024-07-02

**Recommendation:** Accept (Poster)
**Confidence:** 4

**Metareview:**

This submission has been reviewed by four experts, three out of four final ratings from whom are positive after rebuttal. Considering the quality of the submission, the comments from the reviewers, and the rebuttal from the authors, the paper can be accepted by the conference.